# Analysis of the Association between Copy Number Variation and Ventricular Fibrillation in ST-Elevation Acute Myocardial Infarction

**DOI:** 10.3390/ijms25052548

**Published:** 2024-02-22

**Authors:** Roberto Lorente-Bermúdez, Ricardo Pan-Lizcano, Lucía Núñez, Domingo López-Vázquez, Fernando Rebollal-Leal, José Manuel Vázquez-Rodríguez, Manuel Hermida-Prieto

**Affiliations:** 1Grupo de Investigación en Cardiología, Instituto de Investigación Biomédica de A Coruña (INIBIC), Complexo Hospitalario Universitario de A Coruña (CHUAC-SERGAS), GRINCAR-Universidade da Coruña (UDC), 15006 A Coruña, Spain; 2GRINCAR Research Group, Departamento de Ciencias de la Salud, Universidade da Coruña, 15403 A Coruña, Spain; 3Servicio de Cardiología, Instituto de Investigación Biomédica de A Coruña (INIBIC), Complexo Hospitalario Universitario de A Coruña (CHUAC-SERGAS), Universidade da Coruña (UDC), 15006 A Coruña, Spain; 4Centro de Investigación Biomédica en Red Enfermedades Cardiovasculares (CIBERCV), Instituto de Salud Carlos III, 28029 Madrid, Spain

**Keywords:** copy number variants, ventricular fibrillation, acute myocardial infarction, whole exome sequencing

## Abstract

Sudden cardiac death due to ventricular fibrillation (VF) during ST-elevation acute myocardial infarction (STEAMI) significantly contributes to cardiovascular-related deaths. Although VF has been linked to genetic factors, variations in copy number variation (CNV), a significant source of genetic variation, have remained largely unexplored in this context. To address this knowledge gap, this study performed whole exome sequencing analysis on a cohort of 39 patients with STEAMI who experienced VF, aiming to elucidate the role of CNVs in this pathology. The analysis revealed CNVs in the form of duplications in the *PARP2* and *TTC5* genes as well as CNVs in the form of deletions in the *MUC15* and *PPP6R1* genes, which could potentially serve as risk indicators for VF during STEAMI. The analysis also underscores notable CNVs with an average gene copy number equal to or greater than four in *DEFB134*, *FCGR2C*, *GREM1*, *PARM1*, *SCG5*, and *UNC79* genes. These findings provide further insight into the role of CNVs in VF in the context of STEAMI.

## 1. Introduction

Sudden death (SD) is the sudden and nonviolent death of an apparently healthy individual, and it represents one of the most significant challenges in modern cardiology due to its wide incidence and significant social impact [1]. Most cases of SD are typically cardiovascular in nature, and 20% are secondary to ischemic heart disease, mainly due to ST-elevation acute myocardial infarction (STEAMI), especially in patients with ventricular fibrillation (VF) [1,2].

VF rarely spontaneously reverts to non-lethal rhythms, with only 10% to 25% of patients [3] surviving out-of-hospital cardiac arrest. Although this survival rate is increasing significantly, a high mortality rate still remains [4]. Furthermore, in patients who reach the hospital and receive treatment, the presence of VF indicates a short-term unfavorable prognosis, as it has been described that these patients have higher in-hospital mortality [2].

The risk of SD due to ventricular arrhythmias in STEAMI, with a special focus on VF, has been proposed to result from the interaction between genetic factors and environmental factors [1,5,6]. While various genetic variations, such as single nucleotide polymorphisms (SNPs), have been extensively studied in VF associated with STEAMI [7,8,9], there is a notable gap in the exploration of copy number variants (CNVs).

The frequency of CNVs in the human genome is approximately 12% [10,11], making them a common component of the human genome. They play a significant role in evolution, contributing to population diversity [11], but also in the development of certain diseases [12,13] such as cancer [14], autoimmune diseases [15], neurodegenerative diseases [16], and cardiovascular diseases [17] such as tetralogy of Fallot, aortic stenosis, coarctation of the aorta, and ventricular septal defect.

Previous studies have focused on CNVs’ associations with SD [18,19,20,21], but none have been directly related to VF in STEAMI. This study aims to investigate CNVs in the context of STEAMI and VF using NGS technology. The detection of CNVs through NGS is anticipated to provide valuable insights into risk assessment and prognosis, holding significant implications for the prevention and treatment of SD associated with STEAMI.

## 2. Hypothesis and Objectives

The hypothesis is that CNVs significantly influence the predisposition to VF in patients with STEAMI.

Thus, the main objective of this study is to detect and characterize genes within CNVs identified in a cohort of patients with VF in the context of STEAMI. To achieve this objective, we will pursue a dual approach. Firstly, we will examine genes exhibiting CNVs in a significant population of STEAMI patients with VF, and concurrently, we will assess the presence of recurrent CNV patterns within these specific genes. Secondly, we will investigate the average gene copy number within the STEAMI and VF patient population, conducting a comprehensive analysis to identify the specific genes involved.

## 3. Results

The principal characteristics of the 39 patients analyzed with VF during AMI and of the 41 patients without VF during AMI used as control included in the study are summarized in Table 1. There was not a significant difference between the groups in any of these variables.

To ensure the quality of the alignments, a preliminary assessment was performed by analyzing the bioinformatic quality report. This analysis allowed for the verification of the integrity and reliability of the aligned data before proceeding with subsequent stages of the analysis. It is noteworthy that everything in the data was found to be correct during this review.

The complete pipeline was then executed using a single-node configuration with 24 processes, which took approximately 80 min. This execution benefited from parallelization capability through a tool parameter that utilized all available cores on the node, resulting in significant time savings compared to sequential execution.

As a result of this process, a table containing a total of 4688 CNVs in the 41 VF patients was obtained. A preliminary inspection was then conducted to assess the data’s integrity. During this initial review, it was identified that two patients exhibited a significantly higher number of CNVs compared to the rest of the samples. Consequently, a scatter plot was generated to examine the nature of CNVs in these two particular patients. The scatterplots revealed the presence of considerable noise in the CNVs of these individuals, making it challenging to distinguish them from the true CNVs. Although antitarget BED files were calculated to mitigate this issue, the decision was made to exclude these two patients from further analysis to ensure the integrity and reliability of the remaining data. Subsequently, a new analysis was conducted, detecting a total of 1589 CNVs in the 39 VF patients.

### 3.1. CNVs in a Significant Number of Patients

Using the data obtained after the filtering, tables of genes with CNVs in the form of deletions and duplications that occur in a significant number of patients were obtained (see Table 2 and Table 3). It is essential to note that, by what was mentioned in Section 5.5 of this study, a significant minimum threshold of 6 VF patients with specific CNVs has been established. Therefore, the tables presented only include genes in patients equal to or exceeding this threshold.

Regarding the results of the deletions presented in Table 2, the presence of two genes, *MUC16* and *PPP6R1* (Figure 1A), was observed in a significant number of patients, both located on chromosome 19. It is important to note that DisGeNET has associated *MUC16* with heart failure, suggesting greater relevance in the context of this study.

The results of CNVs in the form of duplications in a significant number of patients can be found in Table 3. Duplications were observed in the *PARP2*, *TEP1*, and *TTC5* genes (Figure 1B). It should be noted that all three genes have been associated with heart disease by DisGeNET. *PARP2* has been associated with cardiac hypertrophy, *TEP1* has been associated with myocardial infarction (MI) and ischemic stroke, and *TTC5* has been associated with cardiovascular disease.

### 3.2. Average Copy Number

To achieve the results for the second sub-objective, the average copy number for each CNV associated with different genes was calculated. It is worth noting that the copy number reflects the total number of copies the gene in question has in relation to the control reference, initially estimated at two. The results are detailed in Table 4.

In Table 4, the gene exhibiting the greatest mean copy number is *DEFB134*, with a value of 4.5 copies. The other genes, *FCGR2C*, *GREM1*, *PARM1*, *SCG5*, and *UNC79*, have an average copy number of 4 copies. It is important to note that DisGeNET has associated *DEFB134* with defects in the atrioventricular septum, while the rest of the genes have not been associated with any cardiac diseases by DisGeNET. This underscores the potential relevance of *DEFB134* in the context of this study.

## 4. Discussion

In this study, a comprehensive analysis was conducted to detect CNVs using whole exome sequencing (WES) data from a cohort of 39 patients previously diagnosed with VF in STEAMI. During the investigation, CNVs in the form of deletions were identified within the genes *MUC16* and *PPP6R1*, along with CNVs in the form of duplications in the genes *PARP2*, *TEP1*, and *TTC5*. Furthermore, it was observed that the genes *DEFB134*, *FCGR2C*, *GREM1*, *PARM1*, *SCG5*, and *UNC79* exhibited an average copy number greater than or equal to 4. These 11 genes show significant CNVs that may play a key role in the predisposition and development of VF.

### 4.1. CNV Deletions

It is worth noting that the genes *MUC16* and *PPP6R1* exhibited CNVs in the form of deletions in a significant number of patients, proposing *MUC16* and *PPP6R1* as genes of particular interest in the context of this study (see Figure 2).

In the gene *MUC16*, mucin 16, an association between increased *MUC16* expression and heart failure (HF) has been observed in previous studies [22,23,24]. It is important to note that HF is a factor that can lead to STEAMI and potentially contribute to the development of VF. However, in the present study, a depletion in the copy number of the *MUC16* gene suggests a possible decrease in its expression. Although a reduction in *MUC16* has previously been associated with certain types of cancer [25,26], further studies are needed to elucidate the role of *MUC16* on VF.

In the case of *PPP6R1*, Protein Phosphatase 6 Regulatory Subunit 1, no previous information related to CNVs has been found in the context of cardiac diseases or any other context.

### 4.2. CNV Duplications

It is important to note that *PARP2*, *TEP1*, and *TTC5* were identified as duplications in a significant portion of the patient population (see Figure 2).

The gene *PARP2*, poly(ADP-ribose) polymerase 2, has been associated with cardiac hypertrophy in both mouse and human [27] research studies. These studies have demonstrated that reducing gene expression (knockdown, KO) has a protective effect on cardiomyocytes against cardiac hypertrophy. In the case of *PARP2*, the KO of this gene results in decreased gene expression and, consequently, a reduction in the amount of *PARP2* protein produced, which has a beneficial effect in preventing excessive cardiomyocyte growth [27]. However, when considering duplications of the *PARP2* gene, we may expect the opposite effect. Duplication may lead to an increase in the number of gene copies in the genome, which could result in increased *PARP2* transcription and protein synthesis. Furthermore, there is a well-documented relationship between cardiac hypertrophy and VF [28,29]. Unfortunately, it was not possible to perform an echocardiogram in these patients to determine the degree of hypertrophy in patients with this duplication. However, the effects of this duplication on cardiac function and its association with cardiac hypertrophy may play a crucial role in the development of VF. This reinforces the potential involvement of the *PARP2* gene CNV as a predisposing factor to VF in the context of STEAMI.

Regarding the *TTC5* gene, tetratricopeptide repeat domain 5, a previous study [30] has described that inhibiting *TTC5* expression leads to a significant attenuation of the damage suffered by cardiomyocytes due to oxygen and glucose deprivation, such as in ischemia. Furthermore, overexpression of *TTC5* has been observed to increase oxidative stress and levels of inflammatory factors in cardiomyocytes [30]. These findings suggest that *TTC5* may play an adverse role in the cellular stress response, and its inhibition could confer protective effects at the cardiac cell level. The presence of CNVs in the form of duplications in *TTC5* in this study suggests that this gene may be overexpressed, increasing stress and causing cardiomyocyte injury, which could lead to STEAMI and subsequently VF. Taken together, these findings suggest that *TTC5* could be involved in the underlying mechanisms of VF in the context of STEAMI.

No direct relationship with CNV has been found related to *TEP1*, telomerase-associated protein 1. Instead, associations with SNPs have been reported in cardiovascular diseases. In fact, rs1760898, rs8022805, rs1760897, rs2228041, and rs3093926 were specifically associated with myocardial infarction, while rs2228041, rs2184282, rs2104978, rs8022805, rs1713456, rs1713456, rs1713434, and rs7145318 were linked to ischemic stroke [31].

### 4.3. Average Gene Copy Number

Finally, it is essential to highlight the presence of CNVs with an average gene copy number equal to or greater than four in *DEFB134*, *FCGR2C*, *GREM1*, *PARM1*, *SCG5*, and *UNC79* (see Figure 2).

The gene*DEFB134*, defensin beta 134, has been associated with an atrioventricular septal defect in fetuses [32], and this association has been established in cases involving a complete deletion of *DEFB134*. To date, no reports in the scientific literature have linked the *DEFB134* gene to cardiovascular diseases in adults. This discovery and further research have the potential to elucidate the role of *DEFB134* in VF.

In relation to the genes *FCGR2C*, *GREM1*, *PARM1*, *SCG5*, and *UNC79*, no evidence supporting an association between CNVs and cardiovascular diseases has been identified in the scientific literature. It is important to note that the absence of information in this context does not rule out the possibility that these genes may play a role in cardiovascular health. Additional research is required to provide clarity on the influence of *FCGR2C*, *GREM1*, *PARM1*, *SCG5*, and *UNC79* in the context of VF.

Although some of the statements of this discussion are mainly speculative but based in previous studies, we think that our data open gates to further research in this little-explored field, particularly the study of mechanisms through which the genes described in this study may contribute to develop FV in AMI. One promising option may be the use of CRISPR technology to inhibit the action of the described genes.

## 5. Materials and Methods

### 5.1. Study Population

The study was conducted in patients aged 18 years and older at the Complejo Hospitalario Universitario de A Coruña (CHUAC) who presented with ST-segment elevation myocardial infarction. Two main groups were included in the study population: cases and controls.

The case group consists of 41 patients who experienced SD due to VF in the context of acute coronary syndrome. The control group consisted of 40 patients with STEAMI who did not experience VF.

Exclusion criteria were applied to ensure the homogeneity of the study population. Patients with other known causes of SD, such as cardiomyopathies and channelopathies, were excluded. Additionally, patients whose SD had a cause different from ischemia, such as the abuse of toxic substances that could induce Brugada-like syndrome (like cocaine), were also excluded.

In total, 81 patients diagnosed with STEAMI at CHUAC between August 2017 and January 2021 were included in this study.

### 5.2. Next-Generation Sequencing

Exome sequencing was performed using NGS using a specific panel designed to capture the coding regions. For all 81 patients, the IDT xGen™ Exome Hybridization Panel v2 was utilized. This panel allowed the generation of paired-end reads in which DNA sequences were sequenced from both ends of DNA fragments, thus enhancing the quality and reliability of the obtained results. These sequencing procedures were carried out using the Illumina NextSeq500 platform.

The alignment of the sequences was conducted against the human genome GRCh37/hg19 using BWA-MEM software (Burrows–Wheeler Aligner-Maximal Exact Matches) version 0.7.15.

### 5.3. Storage and Processing of Data

Both data management and a substantial part of the analysis process were performed on the Finisterrae-III supercomputer, located at the facilities of the Galician Supercomputing Center (CESGA, https://www.cesga.es/, accessed on 18 May 2023). This supercomputer provided the computational power and storage capacity required to perform the computationally intensive tasks needed for this study.

### 5.4. CNVkit

To perform the analysis of the CNVs present in the WES data, the CNVkit bioinformatics tool (0.9.10) [33] was implemented. Among the majority of WES CNV detection tools that rely on the RD approach, CNVkit is one of the most comprehensive tools for CNV analysis [13]. The pipeline designed for CNV analysis using CNVkit followed the guidelines suggested in the CNVkit documentation with some modifications.

#### 5.4.1. Calculation of Accessible Coordinates

To optimize and expedite the analysis process, the calculation of accessible genomic regions was performed. These accessible regions are crucial to avoid unnecessary computations, as in most fully sequenced genomes, such as the human genome, there are extensive areas of DNA considered inaccessible for the sequencing process, such as centromeres, telomeres, and highly repetitive regions.

Identifying the coordinates of accessible regions was carried out using the reference genome GRCh37/hg19 and the command --annotate, generating a BED format file that stores only these locations.

#### 5.4.2. Preparation of BED Files for the Target and Antitarget

The BED file for the target contains the genomic regions that are the target for the CNV analysis. The company of the panel provides this file, although it had to be modified to adapt it for CNVkit analysis.

Since exon regions can vary, the --split option was used to split the larger regions so that the average size of the resulting bins approximated the predefined average size of 267 bp. Additionally, because the provided BED file lacked gene labels for each genomic region, the option --annotate was added to include these labels. This involved incorporating the refFlat file obtained from the UCSC Genome Browser for annotation purposes.

The genomic regions for the antitarget were calculated using the antitarget command based on the genomic positions of the target regions. By using the -g option along with the BED file of accessible coordinates from Section 5.4.1, the calculation process was expedited by avoiding read depth measurements in inaccessible areas. The BED file for the antitarget regions provides a background context for CNV analysis in the target regions. When calculating read depth metrics in the antitarget regions, it yields a measure of read depth variability in regions of no interest. This can help differentiate true CNVs from read depth fluctuations throughout the genome.

#### 5.4.3. Read Depth

Using the coverage command, the $log_{2}$ of the mean read depth was calculated using reads from BAM (Binary Alignment Map) files for each patient and the target and antitarget regions. The result is a table that provides the average read depths corresponding to each of the bins defined in Section 5.4.2, centered around the mean read depth of all chromosomes.

#### 5.4.4. Reference

CNVkit requires a control sample group to compare the mean read depths. Control samples should be generated using the same technical protocol to ensure that they exhibit a similar background noise pattern. Due to the experimental design’s nature, control samples from the same patients were not available to create this control group. Instead, genomes from the group without VF were selected as control samples.

For these samples, the reference command was used with the files obtained in Section 5.4.3, along with the reference genome GRCh37/hg19, to create a reference file containing the mean read depths from all control samples. This reference file allows comparison with the remaining samples that present VF.

#### 5.4.5. Normalization and Correction

When the fix command is used, the read depths of the targets and antitargets from the samples are combined, and any bin that does not meet the pre-established standards (in this case, the tool’s default values) is removed. It also corrects systematic biases in bin coverage, subtracts the $log_{2}$ read depths from the reference file, and then centers the median of the ratios to ensure that positive and negative biases are balanced around zero. This step is crucial as it allows real differences to stand out more clearly while reducing the influence of any unwanted noise or variability in the data.

The read depth itself is not a sufficient indicator of copy number due to systematic biases introduced during library preparation and sequencing. For example, read depth is affected by GC content [34], sequence complexity, and interval size.

The fix command can remove most systematic errors by normalizing to a reference. However, even after normalization to a reference, coverage biases often persist compared to individual samples and need to be corrected. To resolve this, three additional correction steps were performed together with the fix command. Here is an explanation of what each command aims to correct:

GC bias correction: GC-rich fragments are underrepresented because DNA regions with a high GC content are less accessible for hybridization and susceptible to amplification during library preparation [35].

Repeated sequences: Repetitive sequences in the genome can complicate read depth calculations because these regions often exhibit high coverage variability from one sample to another [33]. In reference genome sequences, repetitive regions are masked using RepeatMasker (http://repeatmasker.org, accessed on 18 May 2023). CNVkit calculates the proportion of each bin that is masked and uses this information for bias correction.

Target density: There are two systematic read depth distortions that occur at the edges of each target. The boundaries of each target display reduced read depth due to incomplete sequence matching with the probe, creating a negative bias in the observed read depth within the interval near each boundary. Additionally, some captures occur outside the target on the flanks of the baited interval due to the same mechanism. When targets are closely spaced or adjacent, this read depth in the flanks can overlap with a neighboring target, creating a positive bias in its read depth.

#### 5.4.6. Segmentation and Calling

Once all sample bins have been corrected, they can be segmented into regions with a discrete number of copies using the segment command. Before segmentation, the $log_{2}$ values for each bin are automatically filtered to prevent outliers. The segmentation algorithm used, among all options available, is circular binary segmentation [36], as it produced the best results in comparative evaluations conducted by CNVkit creators with exome panels. The segmentation results are presented in BED format.

Finally, the call command directly maps the $log_{2}$ ratios of the segment to absolute copy numbers for genes, based on a set of numerical thresholds.

#### 5.4.7. Genemetrics

The final step of the pipeline involves the identification of specific genes that exhibit alterations in genomic copy number, which exceed a critical threshold. In the context of a diploid genome, gaining a single copy in a sample translates to a copy ratio of 3/2. This relationship can be expressed on a logarithmic base 2 scale, where $log_{2}$ (3/2) equals 0.585. Similarly, losing a single copy is represented as $log_{2}$ (1/2), which is equal to −1.0 and so on.

To carry out this process, the genemetrics command has been used with a specific threshold, following the tool’s documentation recommendations, set at −t 0.4. Additionally, to mitigate the presence of false positives in the results, a filter based on the number of probes per bin has been implemented. Following the documentation, a value of −m 5 has been chosen. Although this choice may exclude some true positives, it significantly reduces the presence of false positives in the analysis, improving the reliability of the final results.

### 5.5. Statistical Analysis

To analyze the demographic characteristics, a $\chi^{2}$ test was performed for the following variables: sex; dyslipidemia; hypertension; diabetes; and tobacco, as these variables were considered dichotomous. For age and BMI, an ANOVA test was performed.

To determine the minimum threshold of VF patients required to consider the frequency of a significant CNV, Fisher’s exact test was used. This test is particularly suitable in situations where the sample size is small and applying $\chi^{2}$ tests is not feasible. The objective is to assess whether the distribution of CNVs presence or absence significantly departs from what would be expected by chance. A contingency table was constructed and a significance level alpha of 0.05 was established. Using R, we computed the minimum sample size required to achieve a significance level below 0.05. A sensitivity analysis indicated that an N ≥ 6 VF patients exhibiting CNVs was indispensable to achieve statistical significance within the context of this study.

In the context of evaluating the average copy number of relevant genes, no specific threshold has been universally established. Previous studies focusing on cancer-related genes [37,38,39] have consistently demonstrated significant variations when gene copy numbers reached or exceeded 4 copies. Consequently, we have opted to implement an average copy number filtration threshold of 4 or greater.

### 5.6. Filtering

Based on the results generated by the pipeline described in Section 5.4, an extra filtering step was implemented to enhance the selection of CNVs. This refinement process was conducted in R, employing the dplyr library (version 1.1.2).

Initially, CNVs with a copy ratio equal to 2, as elaborated in Section 5.4.7, were excluded. This exclusion is rooted in the assumption that CNVs with a copy ratio of 2 do not represent a significant variation when compared to the diploid reference. Additionally, a filtration step was applied to the *p*-values assigned to each CNV, discarding those exceeding the threshold of 0.01. This *p*-value filtration is essential to concentrate the analysis on the most pertinent and statistically significant CNVs. Lastly, CNVs lacking an association with any specific gene were eliminated.

### 5.7. Enrichment through DisGeNET

DisGeNET [40] is a research platform that contains one of the largest public collections of genes and variants associated with human diseases. It was used, along with its R library “disgenet2r” (version 0.99.3), to enrich the analysis by incorporating information about diseases directly related to each gene. It is essential to emphasize that the diseases associated with “disgenet2r” are not limited exclusively to those linked to a particular gene’s CNVs. They encompass a wider spectrum of medical conditions that have previously been related to each gene in earlier research. The purpose of using this library is to identify any previously established link between the gene under study and a disease.

## 6. Limitations

A significant limitation of the study lies in the sample size. However, it is essential to note that obtaining a significant number of samples is a challenge in this context, as VF occurs in less than 3% of MI cases [41]. Another limitation of the study is that the CNVs on the X and Y chromosomes were not analyzed, as it was not the main objective of the study.

## 7. Conclusions

This study has addressed the relationship between CNVs and predisposition to VF in patients who have experienced an MI. We can conclude that: The presence of 3 CNVs in the form of duplications in the *PARP2*, *TEP1*, and *TTC5* genes has been confirmed in 6 patients with VF. The presence of 2 CNVs in the form of deletions in the genes *MUC16* and *PPP6R1* has been confirmed in 10 patients with VF. An association has been observed between the mean copy number of genes *DEFB134*, *FCGR2C*, *GREM1*, *PARM1*, *SCG5*, and *UNC79* and VF in the population of patients with STEAMI. Literature supports the involvement of the *PARP2* and *TTC5* genes as potential candidates of relevance in the pathogenesis of VF in the context of MI.

## Figures and Tables

**Figure 1 ijms-25-02548-f001:**
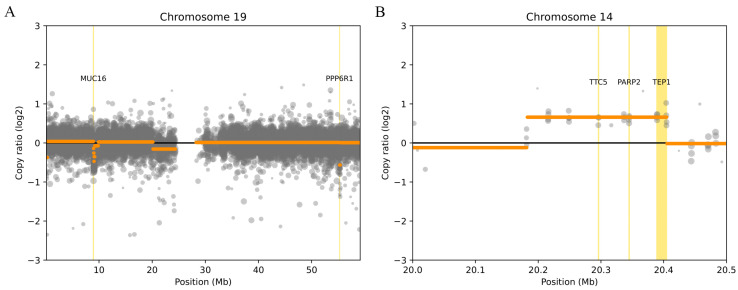
Plot bin-level $log_{2}$ coverages and segmentation calls together. (**A**) Example of deletions on chromosome 19 affecting the *MUC16* and *PPP6R1* genes. (**B**) Example of duplications on chromosome 14 affecting the *TTC5*, *PARP2*, and *TEP1* genes.

**Figure 2 ijms-25-02548-f002:**
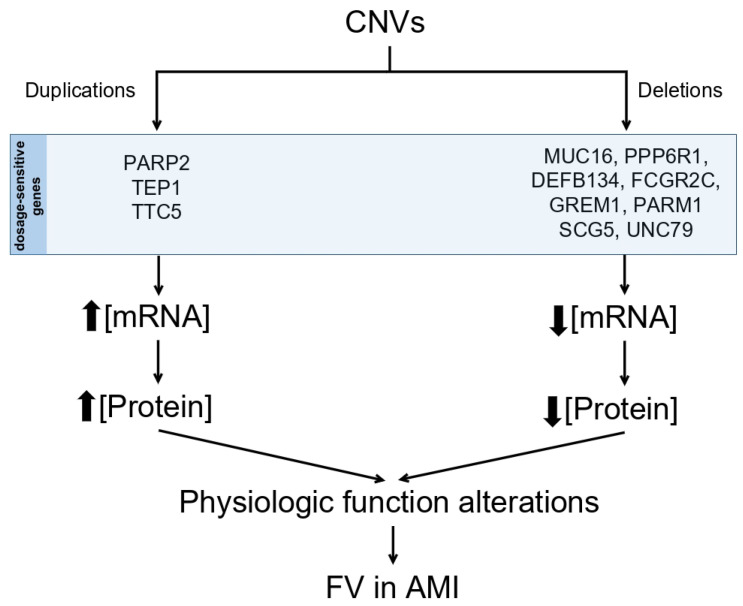
Illustrative potential effect of CNV dosage on FV in AMI.

**Table 1 ijms-25-02548-t001:** Demographic and clinical data of the studied population.

		AMI with VF (*n* = 39)	AMI without VF (*n* = 40)
Age (years)		58.6 ± 11.4	58.5 ± 11.8
Sex (Masculine, %)		78.0	77.5
BMI (kg/m^2^)		28.8 ± 5.0	28.9 ± 4.9
Dyslipidemia (%)		43.6	47.5
Hypertension (%)		43.6	50.0
Diabetes (%)		12.9	17.5
Tobacco (%)		64.1	70.0
AMI Localization (%)	Anterior	51.3	57.5
	Inferior	38.5	37.5
	Lateral	5.1	5.0
	Indeterminate	5.1	-
Vessels affected (%)	1	51.3	50.0
	2	23.1	25.0
	3	25.6	25.0
Time of Ischemia (%)	<120 min	17.9	15.0
	120–360 min	79.5	80.0
	>360 min	2.6	5.0

**Table 2 ijms-25-02548-t002:** Genes with CNVs in the form of deletions in a significant number of patients.

Genes	Chromosome	No. of Patients ^1^	Associated Diseases ^2^
*MUC16*	19	6	Heart failure, Ovarian disease, Stomach carcinoma.
*PPP6R1*	19	6	Prosthetic stomatitis.

^1^ Number of patients. ^2^ Obtained from disgenet2r.

**Table 3 ijms-25-02548-t003:** Genes with CNVs in the form of duplications in a significant number of patients.

Genes	Chromosome	No. of Patients ^1^	Associated Diseases ^2^
*PARP2*	14	10	Cardiac hypertrophy, Breast disease, Cancer.
*TEP1*	14	10	STEAMI, Ischemic Stroke, Campylobacteriosis.
*TTC5*	14	10	Cardiovascular disease, Tubulinopathy.

^1^ Number of patients. ^2^ Obtained from disgenet2r.

**Table 4 ijms-25-02548-t004:** Genes with an average copy number greater than or equal to four.

Genes	Chromosome	MCN ^1^	Associated Diseases ^2^
*DEFB134*	8	4.5	Atrioventricular septal defect, Congenital diaphragmatic hernia.
*FCGR2C*	1	4.0	Autoimmune disease, Thrombocytopenia.
*GREM1*	15	4.0	Cancer, Polydactyly.
*PARM1*	4	4.0	Bladder exstrophy, Cancer.
*SCG5*	15	4.0	Hereditary mixed polyposis syndrome, Cancer.
*UNC79*	14	4.0	CLIFAHDD syndrome, Nicotine dependence.

^1^ Mean copy number. ^2^ Obtained from disgenet2r.

## Data Availability

The data that support the findings of this study are available on justified request to the corresponding author.

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
