# Peer review of "Analysis of the Association between Copy Number Variation and Ventricular Fibrillation in ST-Elevation Acute Myocardial Infarction"

_ijms, 2024, doi:10.3390/ijms25052548_

Round 1

Reviewer 1 Report

Comments and Suggestions for Authors

This study investigates the role of CNVs, in VF during ST elevation acute myocardial infarction. The authors performed whole exome sequencing analysis on 39 and 40 patients with STEAMI with VF and no-VF, respectively. The results revealed duplications in the PARP2, TEP1, and TTC5 genes and deletions in the MUC16 and PPP6R1 genes. Additionally, notable CNVs were found in DEFB134, FCGR2C, GREM1, PARM1, SCG5, and UNC79 genes. This research furthers our understanding of the genetic factors influencing VF in the context of STEAMI. The authors corrected the data for any potential sources of bias or error in their study and laid out the methods well. My specific comments are below.

1. The manuscript lacks a clear definition of what the authors consider a "significant number" of patients. This term is subjective and can lead to misinterpretation of the results. The authors need to clearly define what they mean by "significant" in the context of their study Tables 1 & 2. It would be beneficial to replicate the study with a larger sample size to confirm the findings.

2. Line 62: What do the authors mean by MS?

3. What do the authors mean by ‘abuse of toxic substances’ in line 64? How do they define abuse? What is considered toxic?

4. The authors did not include any information about the demographic characteristics of their study population. Information such as age, sex, and ethnicity could be important confounding factors in this study.

5. The authors' discussion of their results is largely speculative referring to previous exploratory studies. They do not provide any direct evidence to support their interpretations of the data. They need to either provide more evidence or make it clear that their interpretations are hypothetical. For instance, while the study identified CNVs associated with VF, it would be beneficial to investigate the underlying mechanisms to better understand the relationship between CNVs and VF. If the authors can find a CNV associated with VF with high specificity or transcriptional factor that leads to CNV associated with VF or a protein that is associated with the transcriptional factor, that would be impactful, because those can be targeted with CRISPR/cas9.

6. The authors did not discuss how their findings fit into the broader context of existing scientific literature. They need to compare their results to those of previous studies and discuss any similarities or differences.

7. The authors did not provide any recommendations for future research. Given the exploratory nature of their study, it would be helpful to suggest potential directions for future studies.

Author Response

Thank you for your comments. We attach the answer to your question in italic in the text below.

  1. The manuscript lacks a clear definition of what the authors consider a "significant number" of patients. This term is subjective and can lead to misinterpretation of the results. The authors need to clearly define what they mean by "significant" in the context of their study Tables 1 & 2. It would be beneficial to replicate the study with a larger sample size to confirm the findings.

In the 3.5 Statistical analysis are stated the conditions to stablish what a significant number means, related to the number of patients exhibiting CNVs (lines 183-192):

3.5. Statistical analysis

To determine the minimum threshold of VF patients required to consider the frequency of a significant CNV, Fisher’s exact test was used. This test is particularly suitable in situations where the sample size is small, and applying χ2 tests is not feasible. The objective is to assess whether the distribution of CNVs presence or absence significantly departs from what would be expected by chance. A contingency table was constructed and a significance level alpha of 0.05 was established. Using R, we computed the minimum sample size required to achieve a significance level below 0.05. A sensitivity analysis indicated that N 6 VF patients exhibiting CNVs were indispensable to achieve statistical significance within the context of this study.

We agree with the reviewer that “It would be beneficial to replicate the study with a larger sample size to confirm the findings.” Thus, in the 6. Limitations section this idea is already included. The main reason that limits the possibility to replicate the results is due to the low frequency of VF in AMI patients.

  1. Line 62: What do the authors mean by MS?

It has been changed to SD, sudden death.

  1. What do the authors mean by ‘abuse of toxic substances’ in line 64? How do they define abuse? What is considered toxic?

To clarify this point we included: “…abuse of toxic substances that could induce Brugada-like syndrome (like cocaine),…”

  1. The authors did not include any information about the demographic characteristics of their study population. Information such as age, sex, and ethnicity could be important confounding factors in this study.

We have included a new table containing the demographic characteristics of the population (Table 1). Moreover, we have included the following text in the manuscript:

In the stadistics section, we included the following first paragraph (183-184):

“To analyze the demographic characteristics a chi-squared test was performed for the following variables: sex; dyslipidemia; hypertension; diabetes; and tobacco, as these variables were considered dichotomous. For age and BMI, an ANOVA test was performed.”

In the results section, we included the following first paragraph (218-219):

“The principal characteristics of the 39 patients analyzed with VF during AMI and of the 41 patients without VF during AMI used as control, included in the study are summarized in Table 1. There was not a significant difference between the groups in any of these variables.”

  1. The authors' discussion of their results is largely speculative referring to previous exploratory studies. They do not provide any direct evidence to support their interpretations of the data. They need to either provide more evidence or make it clear that their interpretations are hypothetical. For instance, while the study identified CNVs associated with VF, it would be beneficial to investigate the underlying mechanisms to better understand the relationship between CNVs and VF. If the authors can find a CNV associated with VF with high specificity or transcriptional factor that leads to CNV associated with VF or a protein that is associated with the transcriptional factor, that would be impactful, because those can be targeted with CRISPR/cas9.

Thank you for your comment, we agree with you that the interpretation of the data is hypothetical and based on previous published articles in the theme. Unfortunately, after a comprehensive searching in the literature, we could not suggest an underlying mechanisms to help a better understanding of the relationship between CNVS and VF in AMI.  Of course, most research in this field left to be done. Thus, the following paragraph was included in the last part of the discussion (336-337):

“Although some of the statements of this discussion are mainly speculative but based in previous studies, we think that our data open gates to further research in this little explored field. Particularly the study of mechanisms through which the genes described in this study may contribute to develop FV in AMI. One promising option may be the use of CRISP technology to  inhibit the action of the described genes.”

  1. The authors did not discuss how their findings fit into the broader context of existing scientific literature. They need to compare their results to those of previous studies and discuss any similarities or differences.

Unfortunately, we haven´t found other study of VF in AMI that help us to compare our results. However, we have included 11 references to put into context our data in the Discussion section.

  1. The authors did not provide any recommendations for future research. Given the exploratory nature of their study, it would be helpful to suggest potential directions for future studies.

 Answered in point 5.

Reviewer 2 Report

Comments and Suggestions for Authors

Please explain how CVN's corelate with traditional CAD risk factors such as smoking, hypertension, dyslipidemia

The authors do not explain the clinical significance and potential implications of this testing

Authors need to include more figures to explain basics of how CVN's play a role in SD

Comments on the Quality of English Language

Quality of English is good

Author Response

Thank you for your comments. We attach the answer to your question in italic in the text below.

Please explain how CVN's corelate with traditional CAD risk factors such as smoking, hypertension, dyslipidemia

Traditional CAD risk factors are caused by lifestyle as smoking, dyslipidemia, etc…while CNVs are genetically determined and inherited from ancestors. So, no relationship is expected between them. To clarify this point, we have included the demographic and traditional CAD risk factors in Table 1: no differences were found between groups.

The authors do not explain the clinical significance and potential implications of this testing

This study is an association study between CNVs and VF in AMI and as we have included in the discussion section, and we regret to recognize that more research is needed in order to indicate the clinical significance and/or the potential implication of this type of genetic testing in these patients. One main limitation is the low frequency of FV in AMI.

Authors need to include more figures to explain basics of how CVN's play a role in SD

Following your suggestion, a figure with an illustrative potential effect of CNV dosage on FV in AMI was included.

Round 2

Reviewer 2 Report

Comments and Suggestions for Authors

Queries have been satisfactorily revised

Comments on the Quality of English Language

None